# Fossilized Endolithic Microorganisms in Pillow Lavas from the Troodos Ophiolite, Cyprus

**Diana-Thean Carlsson [1,2], Magnus Ivarsson [1,3],\* and Anna Neubeck [4],\***

[1] Department of Paleobiology, Swedish Museum of Natural History, 114 18 Stockholm, Sweden; diana.carlsson@nrm.se

[2] Department of Earth Sciences, University of Hamburg, 20146 Hamburg, Germany

[3] Department of Biology, University of Southern Denmark, 5230 Odense, Denmark

[4] Department of Earth Sciences, Palaeobiology, Uppsala University, 752 36 Uppsala, Sweden

**\*** Correspondence: magnus.ivarsson@nrm.se (M.I.); anna.neubeck@geo.uu.se (A.N.)

**Abstract:** The last decade has revealed the igneous oceanic crust to host a more abundant and diverse biota than previously expected. These underexplored rock-hosted deep ecosystems dominated Earth's biosphere prior to plants colonized land in the Ordovician, thus the fossil record of deep endoliths holds invaluable clues to early life and the work to decrypt them needs to be intensified. Here, we present fossilized microorganisms found in open and sealed pore spaces in pillow lavas from the Troodos Ophiolite (91 Ma) on Cyprus. A fungal interpretation is inferred upon the microorganisms based on characteristic morphological features. Geochemical conditions are reconstructed using data from mineralogy, fluid inclusions and the fossils themselves. Mineralogy indicates at least three hydrothermal events and a continuous increase of temperature and pH. Precipitation of 1) celadonite and saponite together with the microbial introduction was followed by 2) Na and Ca zeolites resulting in clay adherence on the microorganisms as protection, and finally 3) Ca carbonates resulted in final fossilization and preservation of the organisms in-situ. Deciphering the fossil record of the deep subseafloor biosphere is a challenging task, but when successful, can unlock doors to life's cryptic past.

**Keywords:** deep biosphere; fossilized microorganisms; Ophiolite

## 1. Introduction

The igneous oceanic crust has been put forward as the largest potential microbial habitat on Earth [1,2]. However, due to difficulties in sampling live species, this vast biosphere is largely unexplored. Additional to a few successful molecular studies [3–6], paleontological material has been used to study life in the oceanic crust [7,8]. Fossilized microorganisms holds paleobiological information on organismal affinity [9–11], microbe–mineral interactions [12], and clues to early evolution of multicellularity [13]. Microbial life in terrestrial and marine crust is usually referred to as the deep biosphere, which today represents between one-tenth to one-third of all living biomass on Earth while a majority of the rest are land plants [14]. However, before vegetation colonized land (~400 Ma) the deep biosphere was the predominant reservoir for living biomass [14]. Elucidating the fossil record of deep endolithic life is thus crucial to understand life's history on Earth. For example how and when eukaryotes evolved to occupy crucial habitats and how this process influenced the eventual dominance of the total biosphere by multicellular eukaryotes. Investigations of subseafloor paleontological material has further displayed the geobiological role deep microorganisms play, including microbe–mineral interactions like weathering of secondary carbonates and zeolites, or formation of Fe-, Mn-oxides and clays [8,10,12,15]. The connection between microorganisms, element

cycling, and mineralization in these deep niches are essential for the preservation and fossilization of microorganisms but far from understood [8].

The oceanic crust is a heterogeneous environment influenced by plate tectonics, volcanism, sediment overburden, but also characterized by more local anomalies like hydrothermal activity and methane seeps. Recycling and renewal of oceanic crust occur along spreading centers, subduction zones, and seamounts, and it is mainly these areas that are sampled and studied with geomicrobiology in mind [2]. However, the uncertainty in environmental, geological, and geochemical parameters that drives and support deep life are still not well understood. Further investigations are needed to understand the connection between microorganisms and their surrounding geological environment.

This study focus on veins and vesicles in pillow lavas from the 91 Ma Troodos ophiolite on Cyprus [16]. Previous studies has shown the Troodos pillow lavas to host ichnofossils in volcanic glass [17,18] but the current study focus on filamentous body fossils. To understand the past living habitat of the microorganisms, the associated mineralogy and fluid inclusions were investigated. Results indicate a direct relationship between microbial colonization, associated mineralogy, hydrothermal fluids, and fossilization.

*Geological Setting and Sampled Localities*

Cyprus is a part of the Anatolian plate in the Mediterranean Sea. It developed in a supra-subduction zone as an ocean spreading ridge in the Tethys Ocean during the Upper Cretaceous [19]. Crystallization and formation of the ophiolite occurred below the carbonate compensation depth. Active spreading ceased during the mid-Miocene when the oceanic crust was pushed up onto the African plate, creating the Marmonian and Troodos Terrains [20]. Continued movement during the upper Miocene, Pliocene, and Pleistocene contributed to accreted sedimentary sequences, the Circum Troodos Sedimentary and the Keryneia Terrain.

We investigated samples from the Troodos Terrain and the Troodos ophiolite therein. Our focus is on fresh basalt in pillow lavas, far from glassy cooling rims, commonly found in association with this type of setting. The sampling was done in Mathiatis mine, an old open pit in which Au, Ag and Fe was mined up until operations ceased in 1987. Two samples (A and B) were sampled at coordinates Latitude 34 58′31.7, Longitude 33 20′49.3 and Latitude 34 58′31.9, Longitude 33 20′47.0, respectively (Figure 1). The mine stratigraphy involves pyrite-rich quartzite at the bottom of the mine overlain by brecciated and chloritized basalts, followed by a younger lava flow. Sampling for the current study was done on the basalt of the lower pillow lavas at the top. For this study only samples from the pillow interiors were used to exclude later weathering.

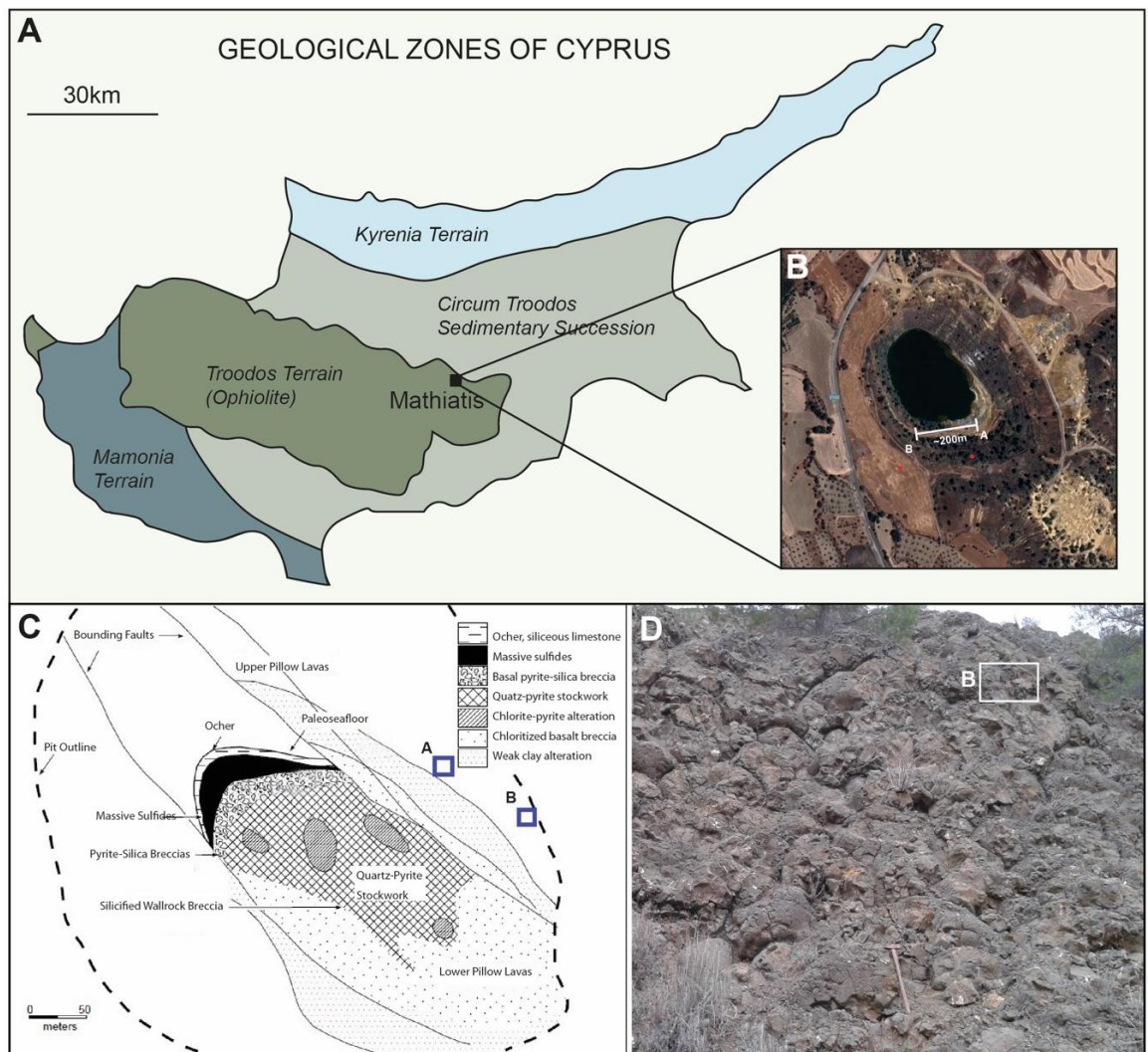

**Figure 1.** (**A**) Map of the geological zones of Cyprus. (**B**) Insert showing sampling locations 07A and 07B. From Google maps. (**C**) Bedrock map of the Mathiatis mine with sampling locations. (**D**) Pillow lavas at the Mathiatis mine with sampling point 07B. Geological hammer for scale.

## 2. Materials and Methods

### 2.1. Microscopy

Thin sections were prepared by Vancouver Petrographics Limited and ABC Ahead in Poland with a thickness of 150–200 μm according to protocols by [21]. Main mineralogy, secondary mineralization, and opaque minerals were analyzed in thin sections using a Nikon Optiphot2 polarization microscope fitted with a Las Ez3 camera. Putative fossilized microorganisms were determined by using a Nikon SMZ1500 microscope equipped with a Nikon D80 camera. Optical microscopy was carried out at the Department of Geological Sciences, Stockholm University, and the Department of Palaeobiology at the Swedish Museum of Natural History in Stockholm.

### 2.2. Environmental Scanning Electron Microscopy

Environmental Scanning Electron microscopy (ESEM) was carried out at the Department of Geological Science at Stockholm University. For this study, a Philips XL-30-ESEM-FEG650 and energy-dispersive spectroscopy (EDS) were used. The electron beam was set to 20 kV and the probe current to 6.00 nA at a working distance of 10 mm. Low vacuum was applied to the sample chamber, giving rise to a conductive atmosphere layer such that carbon coating was not needed for analyses.

## 2.3. Raman Spectroscopy

Raman spectra were collected at the Department of Geological Sciences at Stockholm University, using a confocal laser Raman microspectrometer (Horiba instrument LabRAM HR 800; Horiba Jobin Yvon, Villeneuve d'Ascq, France), equipped with a multichannel air-cooled (−70 °C) 1024 × 256 pixel CCD (charge-coupled device) detector. Spectra were obtained with 1800 lines/mm grating. Excitation was provided by an Ar-ion laser ($\lambda$ = 514 nm) source. Spectra were recorded using a low laser power of 0.1–1 mW at the sample surface to avoid laser-induced alteration of the samples. Analyses were carried out using an Olympus BX41 microscope coupled to the instrument, and the laser beam was focused through 80× (hand specimens, working distance of 8 mm) and 100× (thin sections) objectives to obtain a spot size of about 1 μm. The spectral resolution was 0.3 cm$^{-1}$/pixel, with a typical exposure time of 10 s and with 10 accumulations. The accuracy of the instrument was controlled by repeated use of a silicon wafer calibration standard with a characteristic Raman line at 520.7 cm$^{-1}$. The Raman spectra were achieved with LabSpec 5 software. Minerals in thin sections and hand specimens of collected samples were identified by Laser Raman spectroscopy and comparisons with reference spectra in the RRUFF database [22].

## 2.4. Microthermometry

Microthermometric analyses on fluid inclusions in calcite were performed with a Linkam THM 600 stage mounted on a Nikon microscope utilizing a 40× long-working-distance objective. The working range of the stage is −196 °C to +600 °C [23]. Calibration was made using SynFlinc®synthetic fluid inclusions and well-defined natural inclusions in Alpine quartz. The reproducibility was ±0.1 °C for temperatures below 40 °C and ±0.5 °C for temperatures above 40 °C.

## 3. Results

### 3.1. Mineralogy and Filamentous Structures

The mineralogy of the veins and vesicles was dominated by calcite, zeolites (analcime, mordenite), clays (illite, smectite, montmorillonite) and iron oxides (goethite) according to Raman spectroscopy, see results below. Filaments were found in both open (Figure 2a) and carbonate- and zeolite filled (Figure 2b) veins and vesicles. The latter was studied by thin sections while the first just as hand specimens. They all originate from a clay/iron oxide film lining the interior of the pore space (Figure 3a,b) and protrude from the walls of the host rock into the pore space. The filaments occured either as single features or in abundance, forming complex networks (Figure 2a). They were curvi-linear (Figure 2a), frequently branched (Figure 3c), and occasionally partitioned by septa-like walls (Figure 2b insets). The septation was mostly repetitive and divide the filaments in individual compartments, and in a few filaments a nucleus occured in the center of each compartment (Figure 2b insets). The diameter of the filaments ranged from 5 to 50 μm, and the length ranged from ~50 to several hundred μms. In most cases, the length were reduced by thin section preparation or by damage in open pore spaces. Many filaments also had a central strand that ranged from a few to 15 μm in diameter, making up between a few percent to more than half the diameter of the entire filaments (Figure 3d). The central strand was distinct by its darker brown-red coloration compared to the greyish margins (Figure 3d), and it was possible to see branching of the central strand following the general branching of the filaments (Figure 3e). Occasionally, filaments in open pore space had a terminal precipitation of zeolites forming a swelling at the filament tip (Figure 4).

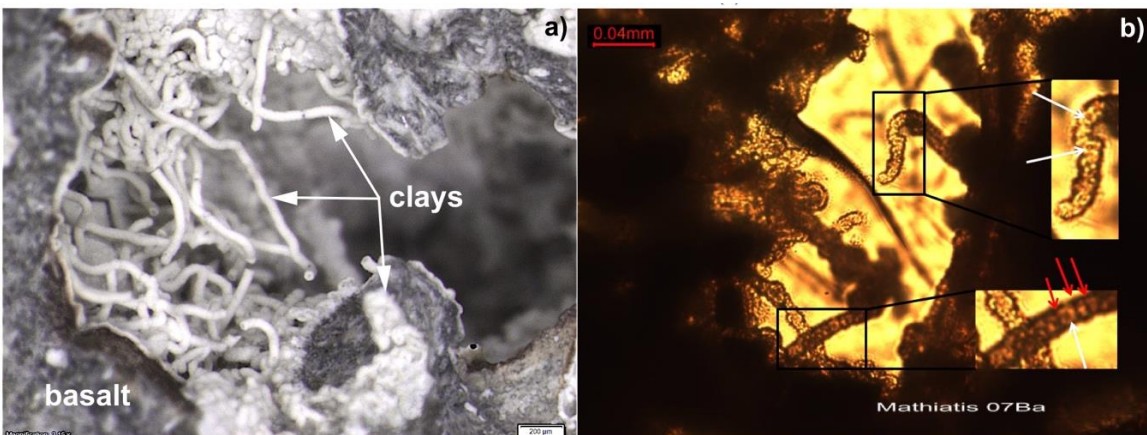

**Figure 2.** Microphotographs of filamentous structures in sample 07A in: (**a**) open vesicles, filamentous structures mineralized by white clay (white arrows) on basalt and; (**b**) calcite filled vein. The filaments protrude from the host rock/pore space interface, are curvi-linear and occur in abundance, forming complex networks. Filaments partitioned by septa-like walls marked by red arrows in 2b (insets). The septation is mostly repetitive and divide the filaments in individual compartments, and in a few filaments a nucleus occur in the center of each compartment marked by white arrows in 2b insets.

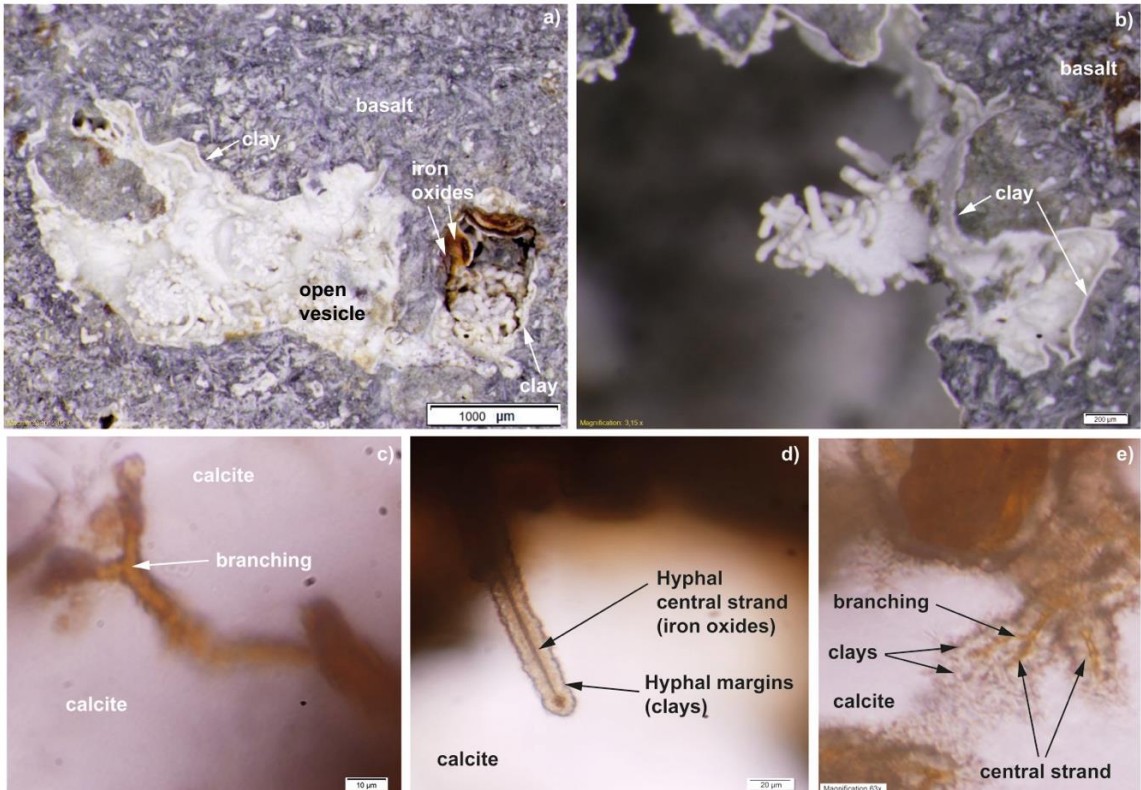

**Figure 3.** Microphotographs from sample 07A showing: (**a**) and (**b**) filaments originating from a clay/iron oxide film lining the interior of the pore space; (**c**) branching of a filament; (**d**) hyphal central strand and hyphal margins; (**e**) branching of the central strand following a general branching of the filaments and mineralizations of clays surrounding the filaments.

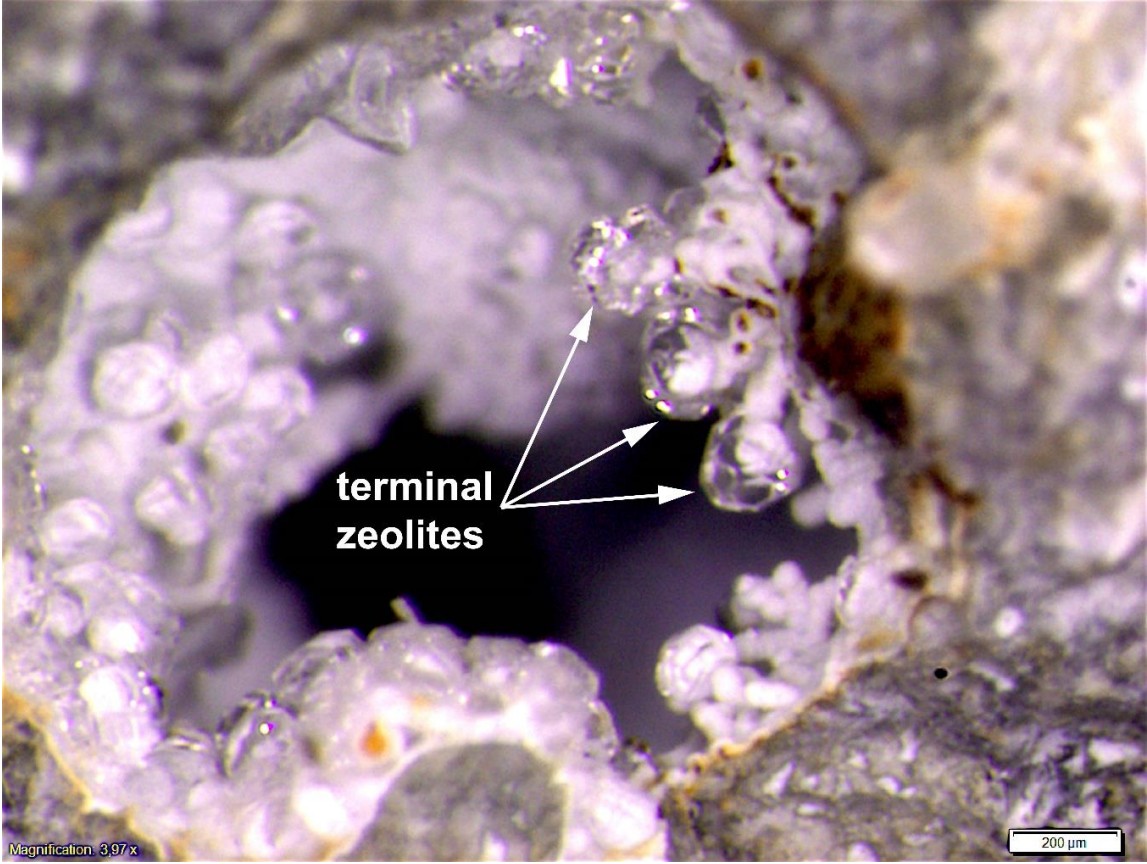

**Figure 4.** Microphotographs from sample 07A of filaments in open pore space with a terminal precipitation of zeolites forming a swelling at the filament tip.

Raman analysis of the film lining the wall interiors, the filament margins and interiors showed discernible peaks at 190, 265, 420 (with one shoulder at 360), 550, 615 and 700 cm$^{-1}$ together with the strong wide band around 3600 cm$^{-1}$ (Figure 5). The wavenumbers are close to published data for typical montmorillonite [(Na, Ca)$_{0.33}$(Al, Mg)$_2$(Si$_4$O$_{10}$)(OH)$_2$*nH$_2$O] with strong bands near 200, 425 (broad band) and 700 cm$^{-1}$ with a less intense peak around 270 cm$^{-1}$ together with a wide band around 3600 cm$^{-1}$ [24,25]. The peaks at 190, 265, 420 and 700 cm$^{-1}$ arise from SiO$_4$ tetrahedral unit vibrations. The peak at 550 cm$^{-1}$ and the downshifted 270 cm$^{-1}$ peak to 265 cm$^{-1}$ indicates an Fe-rich composition, the peak at 420 cm$^{-1}$ is influenced by an Al-rich composition and the peak that occur at the shoulder at 360 cm$^{-1}$ is an indicator for some Mg [25]. The extra band at 615 cm$^{-1}$ is usually not found in montmorillonite spectra, but matches the 604 cm$^{-1}$ peak caused by interference of Fe$^{3+}$ with Si-O-Si groups in the spectrum of glauconite [(K, Na)(Fe$^{3+}$, Al, Mg)$_3$(Si, Al)$_4$O$_{10}$(OH)$_2$] [25]. This extra band may suggest that the Fe-rich nature of the montmorillonite is due to the presence of a glauconite-like structure formed by replacement reactions of the montmorillonite. A mixture of smectite and mica is a common clay association in marine sediments [26] and may also explain why the measured Raman spectra are identified as montmorillonite and not nontronite. Good Raman spectra of clays may be problematic to obtain owing to the ultra-fine texture of the material and low degree of crystallinity [25]. In places, small fragments of rutile appear in the clay (Figure 5). The wide band around 3600 cm$^{-1}$ is assigned to OH vibrations (Figure 5). The appearance of an additional strong peak at 3075 cm$^{-1}$ in the filaments interiors compared to the margins suggest the presence of hydrocarbons and bands in the range 3000–3100 cm$^{-1}$ that can be assigned to C-H stretches in aromatic compounds [27] or =CH$_2$ [28] (Figure 5). The central strand is mineralized by goethite with distinct Raman peaks at 246, 300, 386, 483 and 550 cm$^{-1}$ in the interior of the structures (Figure 6). The presence of hydrocarbons and goethite in the interiors are displayed by a colour variation in optical microscopy. Filaments with compartments

partitioned by repetitive septum-like walls and globular nuclei of 1 or a few µm show a content with a few wt % P according to EDS (Figure 7).

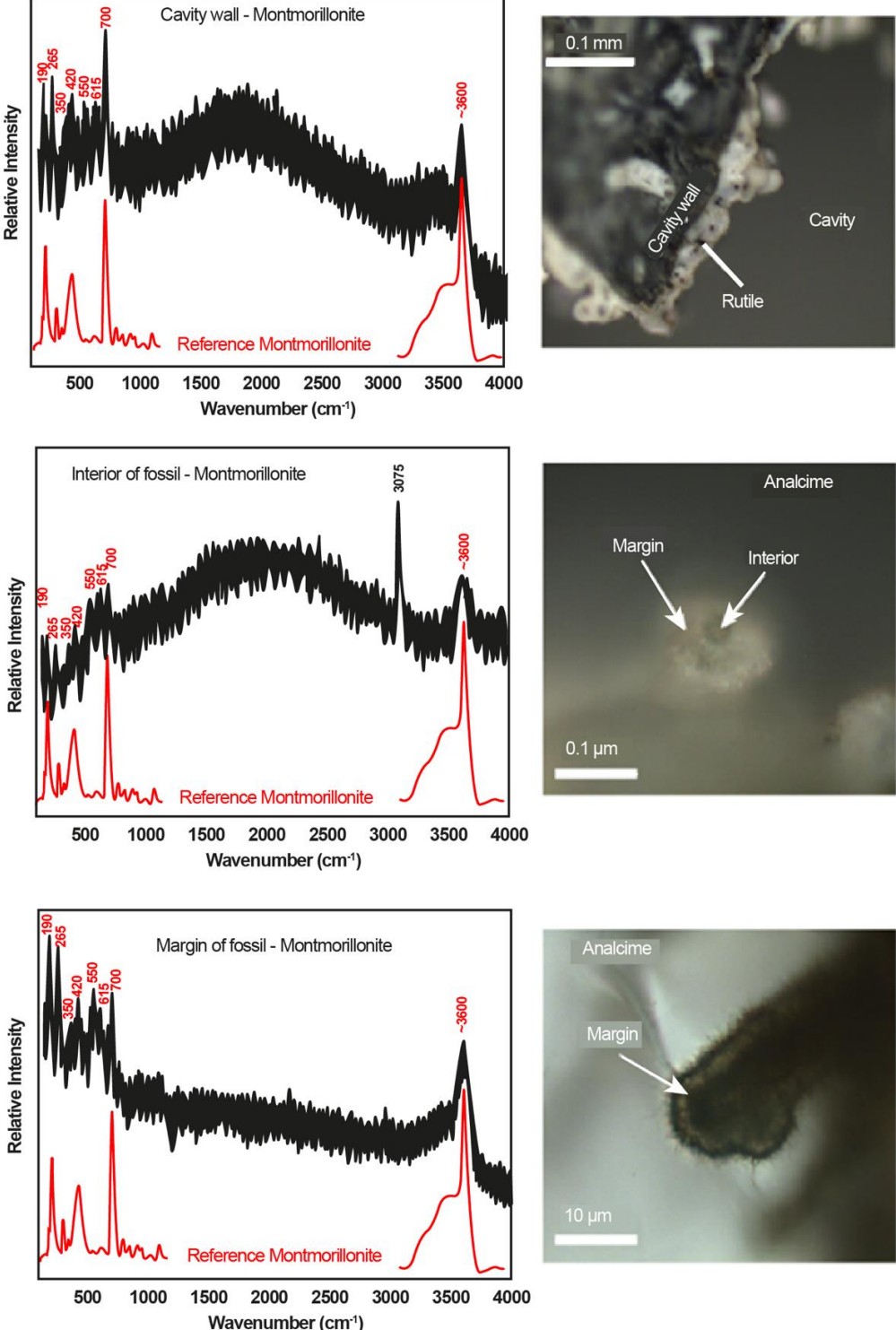

**Figure 5.** Raman spectra in the spectral range 100–4000 cm$^{-1}$ of a cavity wall and filament structure in an analcime-filled vesicle in sample 07A. The spectra with peak positions in red numbers show that both the cavity wall and the filaments are composed of a Fe-rich montmorillonite. The spectra from the interior of the filament indicate that hydrocarbons are present with the peak at 3075 cm$^{-1}$. Reference spectra of montmorillonite after [25] have been incorporated in each spectral diagram. Rutile grains are incorporated in the cavity wall montmorillonite.

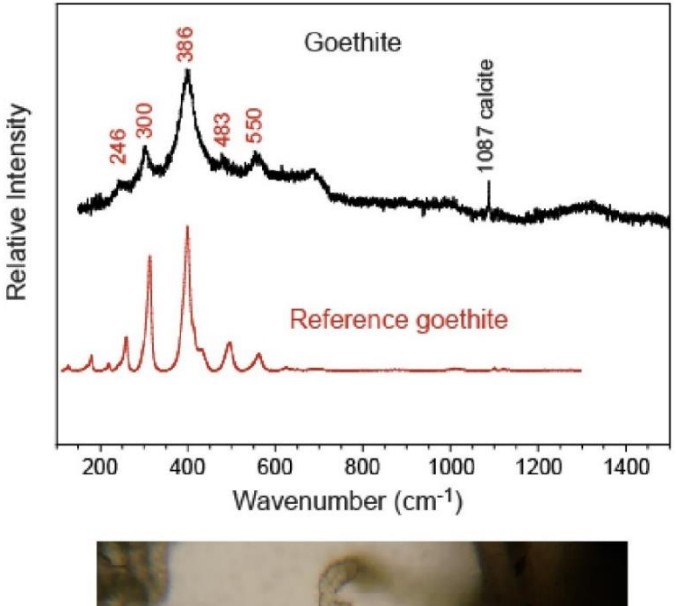

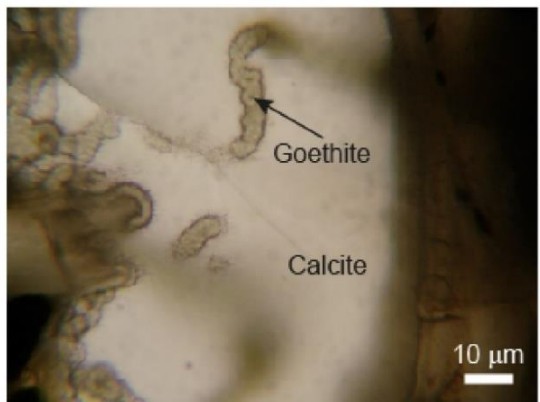

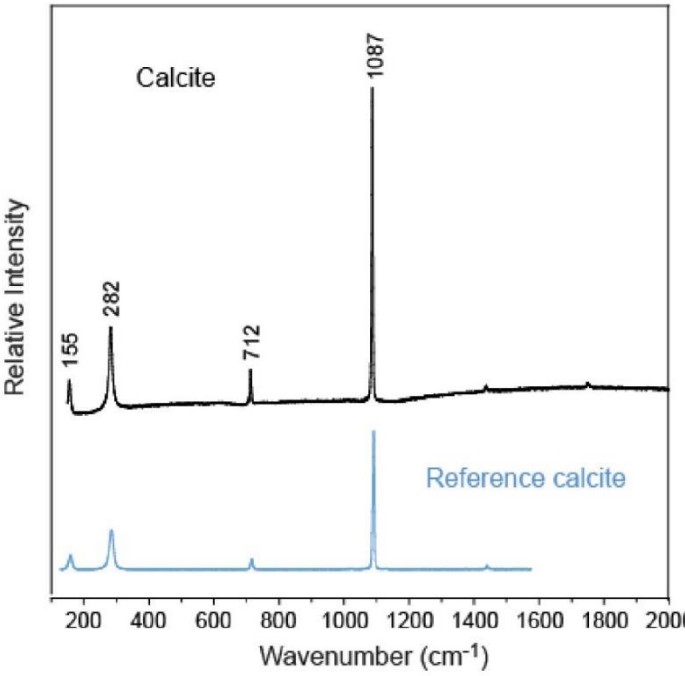

**Figure 6.** Upper diagram (sample 07A): Raman spectrum in the spectral range 100–1500 cm$^{-1}$ of the interior of the filament in a calcite-filled vesicle. The spectrum shows peaks that are attributed to goethite. Lower diagram (sample 07A): Raman spectrum in the spectral range 100–2000 cm$^{-1}$ of a calcite-filled vesicle. Reference spectra of goethite and calcite after Downs (2006) [22] have been incorporated in the figure.

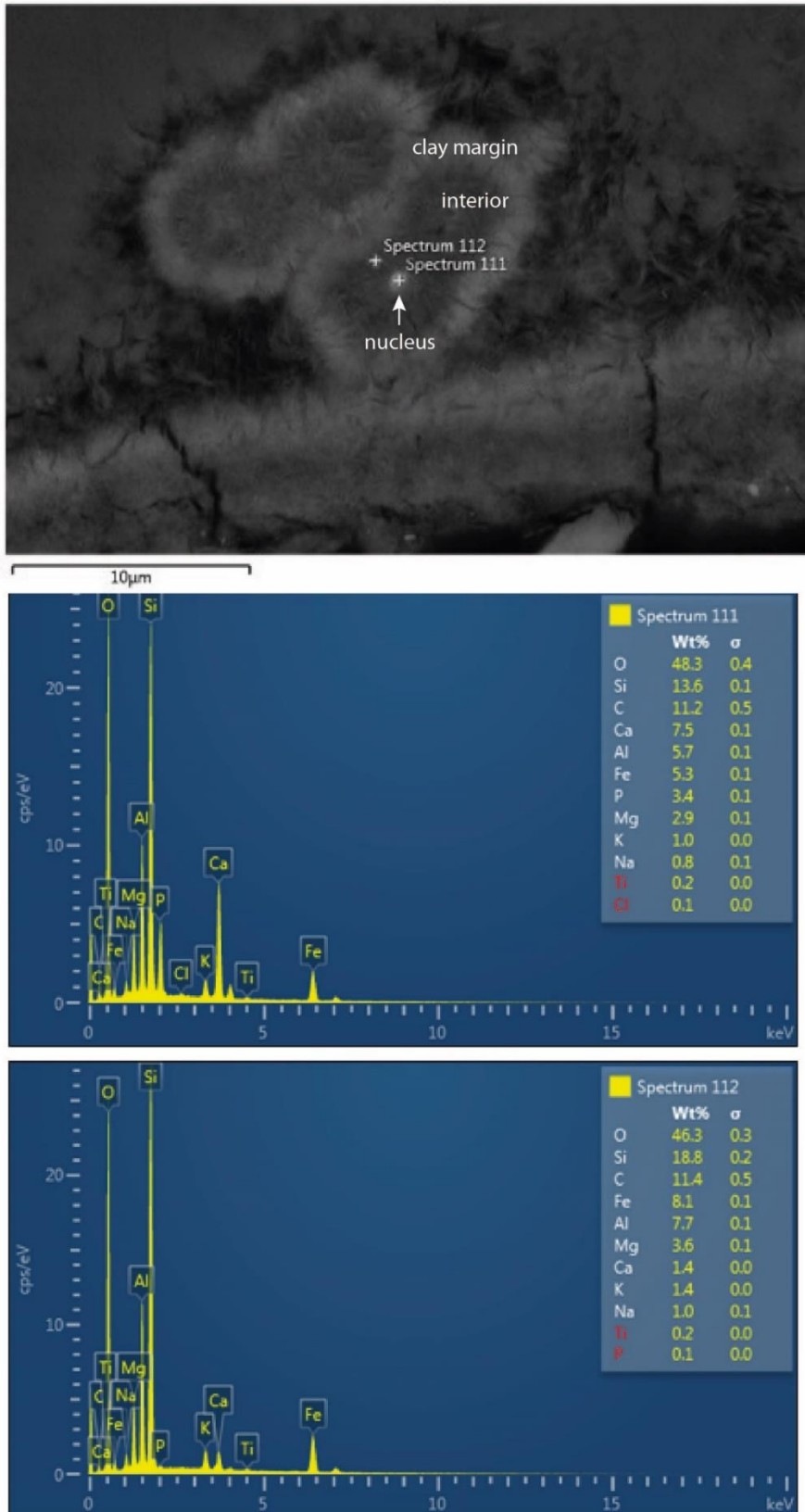

**Figure 7.** ESEM image from sample 07A with accompanying EDS spectra of filaments in cross section showing a phosphorus-rich nucleus surrounded by the overall clay in the filament interior.

Both the film linings and the filaments are seen in association with partially altered orthoclase laths in the vesicles (Figure 8a). The acquired Raman spectra of altered orthoclase (Figure 8b) reflect a gradual transition from orthoclase [$KAlSi_3O_8$] to analcime [$NaAlSi_2O_6*H_2O$]. The strongest Raman bands from the orthoclase that appear at 513 cm$^{-1}$, 475 cm$^{-1}$ with a shoulder at 455 cm$^{-1}$ and 285 cm$^{-1}$ are believed to result from the main T-O-T and O-T-O vibration modes where T = Si and/or Al [29]. Other weaker bands (~755, ~805 and 1124 cm$^{-1}$) are also associated with structural changes in the T-O-T and O-T-O region due to varying bond lengths and angles in the tetrahedral crystal structure [29]. A weaker band at 157 cm$^{-1}$ is assigned to Si-O vibrations involving larger cations like K and Na (Me-O in Figure 8b) in the structure [30]. In spectra A–E (Figure 8b) it can be seen that the replacement by analcime causes the band at 513 cm$^{-1}$ to disappear and the band at 475 cm$^{-1}$ to shift to a higher wavenumber at 482 cm$^{-1}$. The band at 455 cm$^{-1}$ shifts towards a lower wavenumber at 389 cm$^{-1}$ in analcime, whereas the moderately strong band at 285 cm$^{-1}$ for orthoclase shifts upwards to 299 cm$^{-1}$. In analcime, an additional band at 3560 cm$^{-1}$ appear that is assigned to O-H stretching of structurally bonded water [31]. The intensity of this band is gradually increasing from the intermediate phase in spectrum B and absent in the unaltered part (A) of the orthoclase. Areas in the altered orthoclase that have suffered different degrees of transition towards analcime are illustrated in Figure 8b where areas A-B match the corresponding spectra A–E. One larger calcite crystal shows distinct twinning, indicating temperatures <200 °C [32].

*3.2. Fluid Inclusions*

Fluid inclusions are in general rare in the samples but six inclusions in sample 07B with an aqueous liquid and a vapor bubble were distinguished in one specific sample of calcite (Figure 9). Five of these that were found in calcite veinlets, varied in size from 5 to 20 μm and were irregular with angular shapes. Such inclusions may have suffered post-entrapment modifications like stretching or leakage, which can result in a shift to higher homogenization temperatures. The recorded homogenization temperatures from these inclusions, 109 °C to 131 °C (to liquid), should be used with caution and are probably somewhat too high. However, one large fluid inclusion (40 μm) that was found in a cavity filled with well-preserved calcite has survived without leakage. The small size of the cavity and the surrounding basaltic host rock has protected the fluid inclusion from deformation and leakage. Homogenization temperature of this inclusion was measured at 75 °C (to liquid) which is a minimum value for the formation temperature. Initial melting of all six inclusions was observed at temperatures around −35 °C indicating a mixed composition dominated by $Mg^{2+}$, $Fe^{2+}$, $Ca^{2+}$ and $Na^+$ [23]. Final ice melting occurred between −2.4 °C and −2.7 °C, corresponding to salinities from 4.0 to 4.4 eq. mass % NaCl [33].

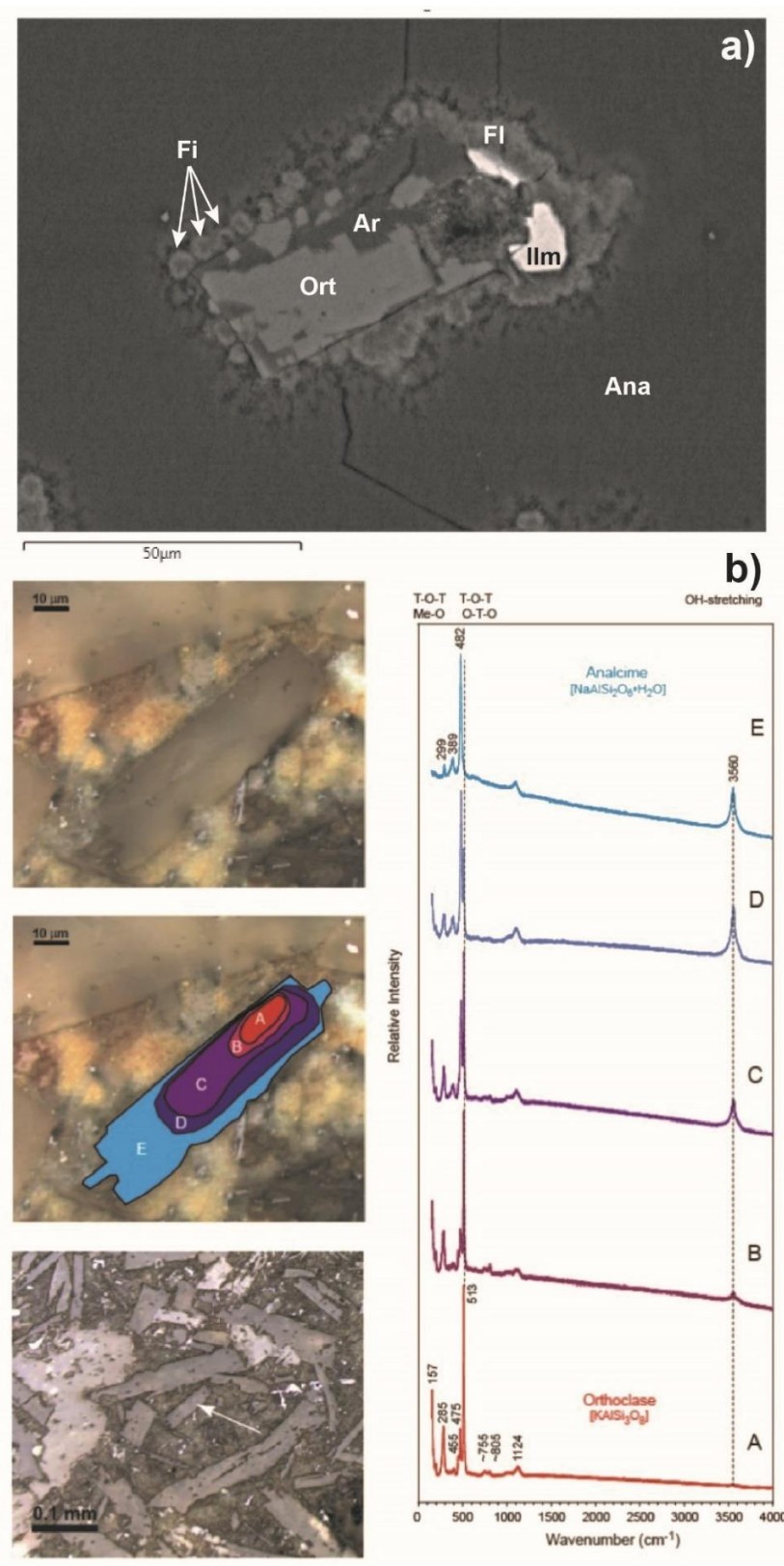

**Figure 8.** (**a**) ESEM image from sample 07A of a orthoclase (Ort) and ilmenite (Ilm) crystal with alteration rims (Ar) and associated filaments (Fi) as well as film lining (Fl) the mineral surfaces embedded in an analcime (Ana); (**b**) Raman spectra in the spectral range 100–4000 cm$^{-1}$ of an altered orthoclase crystal shown in the uppermost microphotograph and its position in the sample (lowermost microphotograph). The Raman spectra A–E illustrate the transition from the endmember feldspar identified as orthoclase (red spectrum A) to the endmember zeolite identified as analcime (blue spectrum E) with intermediate

phases represented by red- to blueviolet spectra B to D. Band positions are indicated by wavenumbers (cm$^{-1}$) on the red spectrum for orthoclase and on the blue spectrum for analcime. The spectral range for vibrational bands Me-O (Me = K, Na), T-O-T and O-T-O (T = Si, Al), and OH is given on top of the diagram. The spectra in the diagram correspond to the areas marked A–E (with the same color) in the middle microphotograph. These areas represent different zones of the gradual transition from the original orthoclase still left in the core to the final phase analcime along the margin of the crystal.

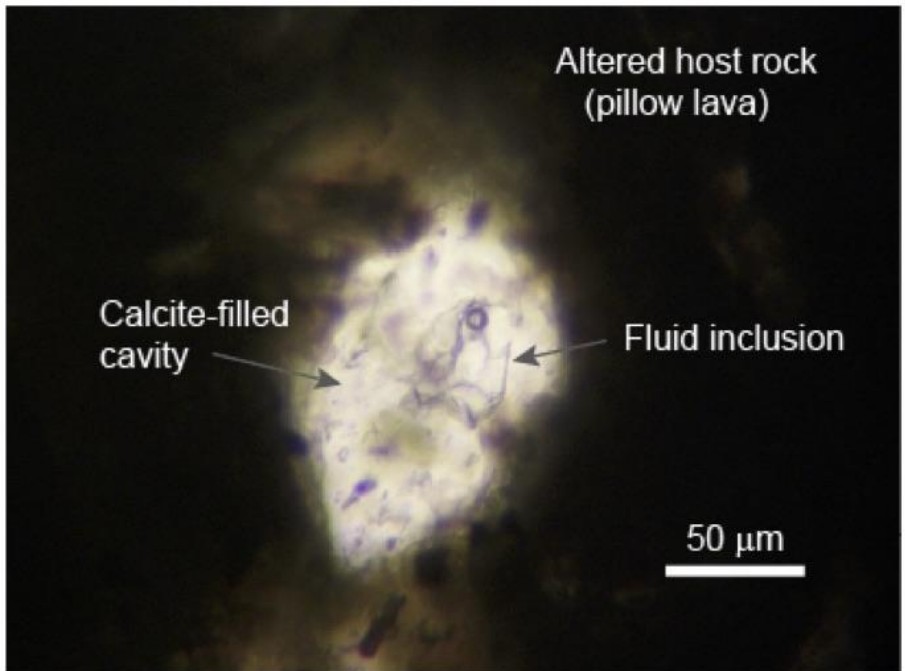

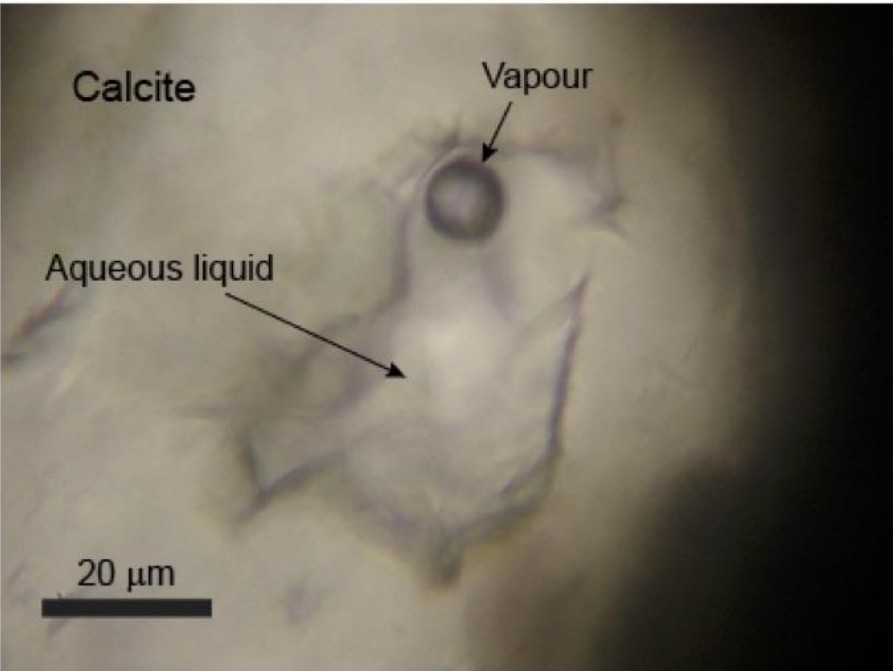

**Figure 9.** Upper panel: Microphotograph from sample 07B of a large aqueous fluid inclusion in a calcite-filled cavity. Lower panel: close up of the same fluid inclusion.

## 4. Discussion

### 4.1. Biogenicity

The following criteria are used to test the biogenicity of the possible biological textures in the pillow lavas: 1) the geological context, 2) the syngenicity and indigenousness to the rock and secondary minerals, 3) the morphology, and 4) the composition of the putative fossils [21,34–36].

1) The filamentous structures are found in both open and filled veins and vesicles in pillow lavas that represent ocean crust of Cretaceous age [19]. Similar filamentous structures has been described from both modern oceanic crust [7,10,12,37] and ophiolites [38,39], and interpreted as fossilized microorganisms. In previous papers, volcanic glass in pillow lavas from Troodos have been shown to host ichnofossils interpreted as morphological remains of microbial activity that occurred prior to the oceanic crust was pushed onto the African plate [17,18]. Furthermore, the oceanic crust is known to host a substantial deep biosphere [12,40], thus, the geological context of the samples is compatible with microbial life.

2) The filaments protrude from the montmorillonite film lining the walls of the pore space and both consist of the same clay, thus the film and the filaments are contemporaneous. No secondary mineralization occurs between the vesicle/vein walls and the suggested microfossils, thus the putative microorganisms colonized the rock during the first alteration stage prior to the formation of the vein-filling carbonates. The filaments are totally mineralized by clays and iron oxides formed by circulating hydrothermal fluids (see Sections 4.2–4.4), and are thus not modern contaminants. Where the filaments are encapsulated by carbonates or zeolites, the secondary mineralization post-dates the growth of the putative microorganisms. Precipitation of secondary minerals and subsequent filling of the voids normally happens 10–20 Ma after formation of the host rock [41]. The ophiolite was emplaced onto land some 20 Ma ago, giving a minimum age of the putative fossils [20] indicating that the microorganisms are syngenetic to early hydrothermal fluids rather than being a modern colonization.

3) No abiotic process is known to form micro-structures with morphologies and growth patterns as those described here. However, there is a substantial literature on microbial fossils in oceanic crust and ophiolites identical to the current structures [12,38] and the references therein. The initial forming of a film lining the interior of the vein/vesicle walls, the subsequent perpendicular growth from the walls into the open pore space, and mineralization by clays and iron oxides is identical to previously described fossilized microorganisms from subseafloor crust and ophiolites, of which most have been interpreted as remains of filamentous endolithic fungi [7,12,37–39,42]. Also, the curvi-linear appearance, the branching, occurrence of a central strand and repeated septations are typical for such filamentous microfossils. No conidia was observed. The width of the filaments ranging between 5 and 50 μm cannot be used as a reliable criterion for biogenicity or to discriminate between prokaryotes or eukaryotes since big bacteria are known to have diameters up to 1 mm [43]. Fungal hyphae are known to have diameters between 2–27 μm, but fossilized hyphae from the ocean floor have been shown to possess wider diameters at times [8,10]. No obvious mineral growth during diagenesis can be seen on the filaments, but due to the thick diameters, it cannot be entirely ruled out. Many of the filaments contains a central strand mineralized by iron oxides and organic matter according to Raman spectroscopy. This could either represent shrinkage of the cytoplasm during fossilization or the remains of thick cell walls [37]. Occasionally, the filaments have repetitive septa resulting in cell-like compartmentalization. Spherical micron-sized textures are found in these compartments, usually containing P (detailed description under point 4). This could also either represent shrinkage of the cytoplasm during fossilization or the possible remains of a cell nucleus. Due to the defined spherical shape, the coherence throughout each compartment, and the exclusive presence of P to the nucleus we suggest the latter. The presence of a viable cell nucleus would exclude a prokaryotic interpretation and be in favor of a eukaryotic interpretation. The filaments appear to have a mycelium-like network, similar to that found among fungi [44], and more or less identical to previously described fossilized fungal mycelium in the igneous oceanic crust [8,10,12]. Actinobacteria are the only known prokaryotes

that form mycelia but have diameters up to 2 μm, thus, are excluded due to the large diameters among the current filaments [45]. In the absence of a known abiotic explanation, and the consistency with a well-known fossil record of the oceanic crust [12], and the references therein, a biological explanation for the morphological characteristics of the current microstructures are the most conceivable.

4) EDS and Raman show that the filaments consist of smectite with a composition of Fe-rich montmorillonite corresponding to fossilization of microorganisms in basaltic crust [12]. EDS also shows P enrichment in possible cell-like interiors, which could be biologically derived. Phosphorous is one of few elements required for all life regarding energy acquirement, genetic information and building membranes [46–48]. Cells containing P are usually consumed by microorganisms during diagenesis. Oxic waters, however, promote preservation of P both through the formation of insoluble P compounds, but also through adsorption and co-precipitation of P onto ferrihydrites [49]. The latter could explain why P has been preserved in the iron-rich central strands.

Some filaments have a carbonaceous content preserved in the central core, according to the Raman analysis (Figure 3). This carbonaceous material is believed to be indigenous to the filaments, and thus represent remains of primary organic matter, which would suggest a biological origin.

The above criteria for biogenicity have been addressed and successfully fulfilled in favor of a biological interpretation. It is further suggested, based on the morphological features that the filaments represent fossilized fungal hyphae rather than filamentous prokaryotes.

## 4.2. Hydrothermal Conditions of the Habitats

Modern hydrothermal vent areas have temperatures between 350 °C–400 °C along the active axial ridge, which decreases gradually from the spreading center [50]. For Troodos, discharge temperatures were between 300 °C–350 °C [51]. One larger calcite from the Mathiatis mine shows distinct twinning, indicating temperatures <200 °C [32]. Homogenization temperatures of fluid inclusions in the calcite from the Mathiatis mine further show temperatures >75 °C. Calcite formation in the sheeted dikes on Cyprus has been estimated to 75 °C–100 °C and is considered to be somewhat higher than calcite formation in the lavas [52]. Zeolite and smectites indicate temperatures <100 °C, and mordenite also indicates temperatures <100 °C [53]. Celadonite is usually formed at temperatures <50 °C, where a previous study has given a formation temperature between 15 °C–20 °C for celadonite on Cyprus, and celadonite found together with saponite in basaltic rocks showed formation temperatures between 50 °C–90 °C [54]. The Raman analysis showed poorly crystalline carbon in the fossilized fungi, which indicates temperatures <150 °C [55–58].

Hydrothermal alteration of the Troodos pillow lavas is given by celadonite and Fe-rich oxides, followed by saponite seen in the vesicles and veins. This stage of alteration should have occurred at temperatures <50 °C for celadonite, and <75 °C for saponite. This sequence is seen in most of the samples, though celadonite is sparse, and saponite is much more abundant. Secondary filling suggests two additional later events with hydrothermal fluids, where the first precipitated Na or Ca zeolites with temperatures <100 °C. The last event precipitated Ca carbonates with temperatures >75 °C.

The organisms thrived in open vesicles prior to secondary precipitation of zeolite or calcite. Thus, this gives a temperature window ranging from ~50 to 75 °C in which the microorganisms could have occurred. The lower range of these temperatures are within the upper temperature maximum for fungi, which is 62 °C [59], while the higher temperature range is above the fungal temperature maximum. This suggests that temperatures partially were within the known temperature limits for fungi during colonization. Furthermore, calcite forms as fluid temperatures increase in the system, and previous studies combining microfossils and fluid inclusions have shown that the temperatures were lower, and within the temperature window for life, in between hotter, mineral-forming pulses of fluids [60].

Basaltic fluids have a general pH between 8–10, but hydrothermal alterations can change the pH in the host rock, where decreasing Mg increases the pH [61,62] Cold oceanic water has a pH between 7–8, which mixes with the magmatic fluids that have a pH between 4–6 when it percolates down into the oceanic floor [63]. During its rise to the ocean floor, pH and/or redox changes will precipitate the

elements in dissolution, creating new minerals. The high clay content in all the samples indicates pH higher than 5 [64]. Mordenite is usually found in environments with a pH between 7–9 [65], analcime is found in environments with pH higher than 8 [66], while Ca carbonates are found in environments with pH higher than 8–9 [67]. Living fungi in aerobic environments have been cultivated and show a preference for pH around 4.5–8.3 [68], and studies of living fungi from subterranean environments shows that they can survive in extreme environments with pH up to 10 [69].

### 4.3. Mobilization of Elements

K and Fe are two of the most abundant cations found in all living organisms. They help control biochemical functions and growth, and the uptake of these ions is thus essential for the microorganisms to survive [70,71]. In the samples, fossilized microorganisms are often found in direct contact with interstitial K-feldspar and Fe oxides (Figure 8b). The Fe-oxides that are in direct contact with the fossils shows no recrystallization along the edges, which indicates either none or Fe-dissolution on a smaller scale than our methods could discern. Recrystallization of K-feldspar at the edges is due to mobilization of $K^+$ and recrystallization with $Na^+$, either abiotically or biologically mediated. The close association with the fossils suggest the latter. Fungi are known to secrete chelating compounds such as siderophores or organic acids to dissolve minerals [72,73]. Biofilm formation is a microbial strategy to initiate microbe–mineral interactions and control micro-environments including surface properties and surface charges [74,75].

The high abundance of fossils and the fact that the samples are from a mine may be due to higher metal content, as well as higher temperatures of the hydrothermal fluids, giving rise to more dissolved ions from the host rock in general. However, the high abundance of putative fungal fossils in the ores, and the connection to a high metal content is probably indirect. Chemoautotrophic prokaryotes favor metal availability and would therefore fix carbon from the $CO_2$ in the fluids, while oxidizing metals that are available in the hydrothermal system. A higher abundance of prokaryotic biomass would mean an increased pool of carbohydrates available for the heterotrophic fungi. Thus, the high fungal abundance in the ore might be a secondary result of a higher chemoautotrophic activity closer to the hydrothermal system. The higher abundance of fungi could also be due to better preservation since more elements for fossilization is available with increasing hydrothermal activity. We believe that with increasing metal content and higher temperature of the hydrothermal fluids (i.e., more dissolved elements), a combination of both these possibilities is likely.

### 4.4. Fossilization

Fossilization of the microorganisms by clays and iron oxides occurred as elements in the fluids successively replaced the organic material. Subsequent mineralization precipitated within or between the cells in the organisms, leaving remains of organic matter in the fossil interiors as revealed by the Raman analysis. There is a known connection between clay minerals and the polymerization of biomolecules, as well as complex biopolymers [72,76–78]. Biomolecules can adhere to vacant sites in clay minerals and there is a molecular structure between the both that favors this coupling. Ivarsson et al. (2013) [7] argued that the fossilization process of subseafloor fungi start while the organisms were still alive, based on the non-dehydrated morphology lacking among dead fungi [79]. During fossilization of the microorganisms in the current study, as well as in previous studies [7,37], this coupling may be of importance in describing how organic molecules can recrystallize into clays. Hydrocarbons contain single bonds that are easy to break. The element-rich fluids that enter the host rock should be able to start recrystallizing the organic molecules by breaking the single bond between the carbon and hydrogen, opening a negative carbon site. Montmorillonite smectite was found as the more common fossilization mineral of the fungi. This is a 2:1 structured clay mineral that consists of 2 repetitive TOT layers. Because of the layered structure in clays, elements are attracted by weak electrostatic bonds making it easy to build the next layer. The TOT structure has an overall negative

charge, which is balanced by inter-layered cations between the TOT layers, usually by cations like K, Na, Ca and water molecules.

EDS and Raman analysis of hyphae in cross-section show that the clay ranges from Mg-rich in the core to Fe- and K-rich in the outer wall, with enrichment of Ti in between. Since K and Fe is thought to have been absorbed by the living organisms, these cations are readily available in situ, while Mg, Ti, Na and Ca most likely have been added by later hydrothermal fluids. All the elements, except Ti, are probably enriched from the seawater, or by the dissolution of the basalt. Mg, however, could also have been dissolved during the breakdown of the clinopyroxene and the olivine in the original host rock. Ti could be probably delivered with the magmatic water and precipitated when pH and temperature changes. Besides, clay and rutile crystals are found in vacant sites of the montmorillonite, and goethite is found in the central strand. The fossils appear to be localized around Fe-oxides, where Fe enrichment is also seen in the clays. Goethite in the fungi might therefore have been due to a high Fe content, and fossilization into an oxide was preferred prior to clay formation, or the fungi might have already deposited the oxide while it was alive. Similar oxide-rich strands have been found in other studies [10,12]. Based on the hydrothermal history, fossilization in these samples might have occurred in the following way:

Microorganisms are introduced into the system with the early low temperature hydrothermal fluids, responsible for the celadonite and saponite alteration. This is followed by a second hydrothermal event with Na, Ca and Si-rich fluids precipitating zeolites, increasing the general pH in the system, as well as the temperatures, and thus stressing the microorganisms to start precipitate clay and oxide minerals as a defense mechanism [7]. The last stage of hydrothermal fluids precipitates Ca carbonates, killing the microorganisms due to too high temperature and pH. Total fossilization probably occurred before total encapsulation, based on fossilized, partly or non-encapsulated microorganisms found in the samples (Figure 2e,f). Hyphae have zeolite precipitated at the tip, which could indicate that the charged surface of these microorganisms might have been a favorable nucleation site. Because the hyphae in the sample show the inner strand in the zeolite encapsulation; this indicates that these microorganisms were encapsulated first after fossilization occurred, since the filament is broken off showing the inner strand. Thus, fossilization seems to be highly dependent on both fluid composition as well as rate and time of secondary precipitation.

## 5. Conclusions

This study focuses on filamentous fossils from the Troodos ophiolite (91 Ma), Cyprus. Based on fungal characteristics such as size, growth behavior, hyphae, frequent branching, central strand, repetitive septa, and potential cell nucleus, a fungal interpretation was inferred for the fossils. Microbial colonization is found on K- and Fe-rich minerals, which they probably adhere to for elemental dissolution. Fossilization of the microorganisms has mainly been done by Fe- and Mg-rich clays, where some fungi have been fossilized by goethite in the central strand, and rutile crystals in vacant sites of the montmorillonite. Elements such as Mg, Ca, and Na have been introduced with the oceanic water, and Ti with the magmatic water. Fossilization was initiated during temperature and pH changes of later hydrothermal activity, where at least three hydrothermal events can be identified. An early hydrothermal event was initiated with temperatures <75 °C, precipitating celadonite and saponite in open vesicles and veins, as well as introduced microbial life into open pore spaces. A second event with temperatures <100 °C precipitated Na and Ca zeolites, increasing the pH to 7–9, stressing the microorganisms into starting to adhere clays as protection. Fossilization was finalized with a last hydrothermal event with temperatures >75 °C, precipitating Ca carbonates, increasing pH to >8–9, as well as making the environment inhabitable for the microorganisms. We, thus, suggest the hydrothermal fluids, temperature, as well as the rate and time of secondary precipitation in open vesicles and veins are the main factors that control preservation.

**Author Contributions:** Conceptualization, D.-T.C., M.I. and A.N.; formal analysis, D.-T.C.; investigation, D.-T.C.; writing—original draft preparation, D.-T.C.; writing—review and editing, M.I. and A.N.; project administration, M.I. and A.N.

**Funding:** This research was funded by the Swedish research council, grant number 2017-04129 and 2013-7320, the Swedish National Space Agency, grant number 100/13, and a Villum Investigator Grant to Don Canfield, grant number 16518.

**Acknowledgments:** The authors acknowledge Marianne Ahlbom, Stockholm University for assistance with the EDS and SEM analyses, and Curt Broman, Stockholm University, for Raman spectroscopy and fluid inclusion analyses.

**Conflicts of Interest:** The authors declare no conflict of interest.

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
