# Peer review of "Fossilized Endolithic Microorganisms in Pillow Lavas from the Troodos Ophiolite, Cyprus"

_geosciences, doi:10.3390/geosciences9110456_

Round 1

Reviewer 1 Report

This is a careful study centered on fungal filaments (hyphae) from Troodos Cretaceous pillow lavas.

In the Abstract and the Introduction the authors often refer to the deep biosphere and I have some doubts on its use. It seems to me that the meaning of "deep biosphere" is confused with the meaning of deep marine biosphere. These are two different things and it is necessary to distinguish: this paper is not devoted to the deep biosphere because the pillow lavas are superficial submarine environments.

The paper refers to previous studies carried out on the biota of these pillow lavas, however the authors neglected to consider the careful microbial study by Furnes et al., 2007 (Precambrian Res. 158, 156-176) that also includes the Troodos pillow lavas and should be taken into account.

There is a poor description of the physical context of the sampled points. It would be better to implement it a little with more explanation and, if possible, pictures.

Minor remarks:

Line 25: keywords need to be corrected

Author Response

Reviewer 1.

In the Abstract and the Introduction the authors often refer to the deep biosphere and I have some doubts on its use. It seems to me that the meaning of "deep biosphere" is confused with the meaning of deep marine biosphere. These are two different things and it is necessary to distinguish: this paper is not devoted to the deep biosphere because the pillow lavas are superficial submarine environments.

We agree that the term “the deep biosphere” is sometimes problematic and there are just as many definitions on what the deep biosphere includes as there are researchers working on it. Therefore, we have rephrased the parts of the abstract and introduction where the term “the deep biosphere is used. We have rephrased to “deep life” or “life in the oceanic crust”.

The paper refers to previous studies carried out on the biota of these pillow lavas, however the authors neglected to consider the careful microbial study by Furnes et al., 2007 (Precambrian Res. 158, 156-176) that also includes the Troodos pillow lavas and should be taken into account.

Furnes et al., 2007 is added in the Introduction and the Discussion.

There is a poor description of the physical context of the sampled points. It would be better to implement it a little with more explanation and, if possible, pictures.

We have added two images in Figure 1 to show the bedrock geology in the mine as well as a photograph of the sampled pillow lavas. We believe this addition address the reviewers concerns.

Minor remarks:

Line 25: keywords need to be corrected

Author response: The keywords are corrected.

Reviewer 2 Report

This paper needs to be rethought.  You are analyzing living organisms living in voids in basalt in an open mine that gets rainwater.  I didn't bother to critically read discussion, because this is where you go off by not understanding this.  I'm going to suggest a few papers to read to help you refocus and rewrite discussion.  Your finding of analcime precipitated by the bacteria is stunning, and worthwhile saving this paper.

Mostly, you are looking at iron bacteria (Fig. 6 is probably Leptothrix cholodnii).  The fact that you found hydrocarbons and aromatics should have triggered your thoughts that maybe these were living.  I understand the interest in fungi, but fungi are heterotrophs and they need organic carbon.  If you go into a mine having rotting timbers, you can expect to find fungi.  Your moist voids in surficial pillow lava might have a few fungi if the moisture is sufficient to support a population of bacteria.  Using your techniques you couldn't possibly see a nucleus.

When this is rewritten, come back and I'll fix the little things (verb tense, Upper Cretaceous, etc.).

You might consider doing XRD and separating out ferrihydrite from goethite (scan from 5-80 2-theta at 1 degree/min).  The iron bacteria precipitate ferrihydrite that reforms into goethite if the water gets dysoxic.

Papers: 1) McAllister, Moore, etc, 2019, FEMS Microbio. Ecol. 95 on iron bacteria at the Loihi seamount--that will get you into the other papers by Emerson. 

2) Robbins, Papadeli, etc, 2016 Geomicro Jour. 33, for the ferrihydrite goethite transition

3) Robbins, Rodgers, etc, 2000, Hydrobiologia 433, for organisms living in mines; and Fig.2-7 for mineralized filaments

Author Response

Reviewer 2

This paper needs to be rethought. You are analyzing living organisms living in voids in basalt in an open mine that gets rainwater. We respect the opinion of the reviewer but it is clear that the reviewer don`t understand the geological context and history of our samples. The reviewer believes the samples represent modern biology, however, there is nothing in the samples supporting the presence of modern biology. All evidence points in the direction of fossilized microorganisms. The comment by the reviewer is not grounded in the evidence and we urge the reviewer to support his/her statement. Our evidence for fossilized life are as follows:

The biological structures are found in petrographic thin sections prepared from the inner parts of the pillow lavas. Total mineralization of the biological structures by minerals corresponding to marine conditions. The mode of mineralization characterized by clays and an occasional minor central strand of iron oxides corresponds perfectly to submarine fungal fossils rather than remains of iron oxidizing bacteria. The latter is only known in the literature to be mineralized and fossilized by iron oxides, never by clays. The fossilized microorganisms are embedded in carbonates and zeolites formed by seawater evidenced by fluid inclusions showing temperatures (75-130°C) and composition for seawater (4.0-4.4 eq. mass % NaCl) and hydrothermal fluids. The temperatures and composition exclude rainwater as a possibility. Thus, the microorganisms were entombed submarine, prior to the relocation of the volcanic section onto land. The mineral succession shows that the microorganisms undoubtedly were mineralized and entombed on the seafloor and not in the present mine environment. Celadonite together with saponite in basaltic rocks show formation temperatures between 50-90°C, indicating a hydrothermal origin, thus, once again indicating a submarine origin of the minerals. Raman analyses of the carbonaceous matter indicates that the hydrocarbons are poorly crystalline, thus, decomposed, and have been subjected to elevated temperatures. Thus, the hydrocarbons can not represent modern biology.

   All these evidence is clearly stated in the Result section and in the Discussion section, which the reviewer claimed to neglect to read!

I didn't bother to critically read discussion, because this is where you go off by not understanding this. The duty of a reviewer is to read the entire manuscript. Neglecting to fulfil this duty has obviously lead to unnecessary misinterpretations of our manuscript.

I'm going to suggest a few papers to read to help you refocus and rewrite discussion.

The suggested papers only describes living iron-oxidizing bacteria, which is of no relevance for our study whatsoever. See comments above on differences in mineralization between iron oxidizing bacteria and fungi as well as the fact that our samples represent fossil material and not living.

Your finding of analcime precipitated by the bacteria is stunning, and worthwhile saving this paper.

Maybe if the living organisms had formed the zeolite. In our samples the fossilized hyphae have only worked as nucleation for mineral growth as described in the Discussion section.

Mostly, you are looking at iron bacteria (Fig. 6 is probably Leptothrix cholodnii). The fact that you found hydrocarbons and aromatics should have triggered your thoughts that maybe these were living. I understand the interest in fungi, but fungi are heterotrophs and they need organic carbon. If you go into a mine having rotting timbers, you can expect to find fungi. Your moist voids in surficial pillow lava might have a few fungi if the moisture is sufficient to support a population of bacteria.

See comments above. The fossils are not iron oxidizing bacteria, and the environment in which they once lived was the subseafloor pillow lavas, not the present mine. All evidence support this. In our opinion figure 6 show no similarities to Leptothrix cholodnii. The “coiled” structure is due to cutting of the filament during thin section preparation, and goethite arrow only shows the minor central strand. About 1 percent of the filament consist goethite, the rest is clays, so not what you expect from a iron oxidizing bacteria.

The presence of hydrocarbons is by no means an indication for living organisms. Examples of hydrocarbons from non-living systems are oil, gas, kerogen, coal etc. Hydrocarbons in fossilized material is the most certain criteria for biogenicity of fossils. The raman peaks of our material shows that it is poorly crystalline and decomposed. If the raman analyses had been performed on living material we would have seen numerous peaks in the spectral range between 600 and 2000 cm-1, see Huang et al., 2004, Analytical Chemistry 76.

Using your techniques you couldn't possibly see a nucleus.

Both optical microscopy and ESEM have high enough resolution to detect a cell nucleus. We have used 100x magnification in the optical microscopy, which enables a resolution of 0.2 microns. ESEM enables a resolution on the nano meter scale.

When this is rewritten, come back and I'll fix the little things (verb tense, Upper Cretaceous, etc.).

You might consider doing XRD and separating out ferrihydrite from goethite (scan from 5-80 2-theta at 1 degree/min). The iron bacteria precipitate ferrihydrite that reforms into goethite if the water gets dysoxic.

Raman spectroscopy is a fine method to separate iron oxides from each other. There is a substantial difference between the goethite and ferrihydrite raman spectra, see reference by Hanesch, M. 2009 Geophysical Journal International vol 177 (3), 941-948.

As we have stated above, the mineralization support a fungal affinity rather than a bacterial affinity. Goethite is only found in the central strand, otherwise the biological structures are mineralized as clays. The growth behaviour also corresponds to fungi rather than iron oxidizing bacteria. Please see the Discussion.

Papers: 1) McAllister, Moore, etc, 2019, FEMS Microbio. Ecol. 95 on iron bacteria at the Loihi seamount--that will get you into the other papers by Emerson.

2) Robbins, Papadeli, etc, 2016 Geomicro Jour. 33, for the ferrihydrite goethite transition

3) Robbins, Rodgers, etc, 2000, Hydrobiologia 433, for organisms living in mines; and Fig.2-7 for mineralized filaments

Round 2

Reviewer 2 Report

line 30 could be (third person--is suggested--is weak)

38 was probably the predominant (we have no data)

39 For example (start a new sentence here, as written it runs on)

47 influenced by abiotic processes (shows that you are paying attention to both)

64 Upper Cretaceous (I told you this before that upper is capitalized as a time term)

69 start We Investigated as a new paragraph

72 Tell us they mined Au, Ag, Fe

78 What is the white substance in the open void?  I think I finally figured it out in line 178, analcime?  Tell us in the Figure.

78 These rocks look highly weathered to me--maybe it is just the photo, because online I found some images of unweathered pillows there.  I wish you would address this issue--that you chose your samples from unweathered rocks.

79 Do I understand correctly that you just had two samples?  My agency would never let me publish a paper on two samples (unless they came from the moon).  Please address this with an explanation as to why you just had two samples.  Did you have others and found nothing useful?  Do you suppose if you looked at another 8 samples you might have found more evidence for the organic carbon source for your putative fungi?

93 were used (I think the verb tense is plural; is the EDS on the Phillips, then it should say with EDS, or if these are two different instruments than plural is correct

118-136 I recommend using past tense for this entire paragraph; these were your findings.

122 The latter confused me; I had to read this over and over.  I think you are saying the filled ones were thin sectioned.  The open voids still sound to me like you can't really be sure when they were colonized.  I wish you would address this somewhere in the article.

128 I still disagree that you are looking at nuclei of fossilized microorganisms.  I wish you would address this somewhere in the article.

137 These look like actinomycetes to me.  And they look modern. I wish you would address this issue of intact organisms in voids

147 I see rods in photo e.  I recommend stating this so that others can focus on these microbial structures.

151 show (not shows)

154 fix formula (upper case H2O)

167 occur (not appear)

174 spell out one

179 an (not a) Fe-rich

181 better to say each spectral diagram

204 take off  parentheses

253 Furthermore (besides is slang)

264 You didn't address post-depositional colonization

290 large (not exceeding)

299 There are lots of phosphatic minerals in the rock record.  Take off that sentence that begins "As".

300 are (not is)

302 This is the first time you bring up ferrihydrite.  If you are thinking about ferrihydrite, then you should also be thinking about iron bacteria.

303 has been (not have)

309 The lack of conidia should at least be addressed.  Maybe if you had more samples, you might have found conidia and made your case stronger.

335 Furthermore (not besides).  Caliche forms (not form). Calcite is also precipitated by organisms.

356 Rewrite this sentence (than our methods could discern)

364 putative fungal fossils

369 fungal (not fungi)

376 as ions in the.  Fossilization of the organisms by clays, xeolites, and Fe oxides

378 as revealed by (not according to)

382 ue past tense for this entire sentence

400 Ti could be (not is)

404 prior to clay formation

406 (take off by)

408 I could write another scenario: Pillow extrusion, sea water chemical interactions produce the  iron oxide and clay substrates.  Bacteria introduced with sea water, colonized later by fungi.  I know you worked hard on 408-420, but I recommend eliminating most of this primarily because you only looked at two samples.

423 Add the Le Calvez et al., 2009, Fungal diversity Appl Env Micro paper

424 for (not on)

425 found on (not around)
